# Perspectives on health, illness, disease and management approaches among Baganda traditional spiritual healers in Central Uganda

Yahaya H. K. Sekagya[1,2]*, Charles Muchunguzi[3], Payyappallimana Unnikrishnan[4], Edgar M. Mulogo[5]

**1** Department of Pharmacy, Mbarara University of Science and Technology, Mbarara, Uganda, **2** Research and Training Department, Dr. Sekagya Institute of Traditional Medicine, Mbarara, Uganda, **3** Department of Environment and Livelihoods Support Systems, Mbarara University of Science and Technology, Mbarara, Uganda, **4** University of Transdisciplinary Health Sciences and Technology, Bengaluru, India, **5** Department of Community Health, Mbarara University of Science and Technology, Mbarara, Uganda

* ysekagya@gmail.com

**Data Availability Statement:** All relevant data are included within the paper.

## Abstract

In Uganda, spirituality is closely associated with traditional healthcare; however, though prevalent, it is considered controversial, mystical, less documented and often misunderstood. There is a paucity of literature on the description of health, illness, disease, and management approaches among traditional spiritual healers. This article examines the perspectives on health, illness, disease, and management approaches among Baganda traditional spiritual healers, the *Balubaale*, in Central Uganda, who engage ancestral spirits during health care and management. We used a qualitative study design in particular grounded theory. We used semi-structured, qualitative interviews and observation on 12 male and female purposively selected *Balubaale* in Central Uganda. Data was transcribed, coded, and thematically analyzed using ATLAS ti. 22 Computer software based on an inductive approach. Findings show that the words and concepts describing health, illness, disease, and management approaches are descriptive and contextualized to include the problem, the prospected root-causes, and the therapeutic approaches involved. The words for illness "*olumbe*", disease "*obulwadde*" and the management approaches such as divination (*kulagula*), ritual cleansing *(kwambulula)*, amulets (*ensiriba* and *yirizi)*, and scarification (*kusandaga)* have spiritual and social dimensions, contextual meanings and attachments. Further research is recommended among other tribes and larger sample size to compare findings and terminologies to facilitate communication and policy considerations.

## Introduction

Traditional spiritual healers are Indigenous healers who believe in spiritual guides and use them during health management [1, 2]. The worldviews of traditional spiritual healers regarding health, illness, disease, and their management approaches are rooted in their local oral cultures, and traditions, and are associated with witchcraft [3]. During the colonial period, Western civilization and European explorers described Indigenous people as primitive and rendered their Indigenous knowledge systems superstitious, obsolete, irrelevant, and worthless

**Funding:** This work was supported by The Africa Centre of Excellence (ACEII) for Pharm-Biotechnology and Traditional Medicine (PHARMBIOTRAC) under Mbarara University of Science and Technology, Mbarara, Uganda (P151847) to YHKS. The funders did not play any role in the study design, data collection, analysis, decision to publish, or preparation of the manuscript. No relevant grant or award recipients are specifically associated with the funding received for this study.

**Competing interests:** The authors have declared that no competing interests exist.

[4]. In addition, the introduction of education curricula banned people from using their local languages, cultures, rituals, and all spiritual things, and numerous punitive laws and regulations were instituted [5, 6] by the colonizers. These are some of the primary reasons for the little research on this subject [7]. Hoppers and Sandgren suggested an equitable dialogue between academy and Indigenous knowledge holders [8], which should inform the social contracts that underpin the different understandings of knowledge within the diverse settings of global societies [9]. In 2014, Bartel documented that true knowledge and wisdom are better expressed and communicated as first-hand personal experiences in the vernacular knowledge of the people's places [10]. Health management in the context of people's traditions and customs is encouraged since its existence determines the basis for its acceptance, criticism, or rejection [11, 12].

Even though the scientific disciplines of sociology and philosophy of medicine have their concepts of health, illness, and disease [13, 14], it is not rational/valid to challenge indigenous research to uncover the traditional logic and methods about such concepts with mainstream scientific equivalents. Instead, the emphasis should be to continue research on traditional knowledge without undermining its knowledge or reducing its holistic cultural values [15, 16]. However, the challenge remains for policy formulation where the use of local descriptions for health, illness, disease, and health management may reflect divergent meanings in contemporary science [17–19].

Although, nearly 80% of the population in developing countries uses traditional medicine [11, 20], the Eurocentric science does not acknowledge Indigenous spiritual healers and their use of local descriptive words during health management [7, 21]. However, there are reported global efforts to revitalize Indigenous health systems towards achievement of sustainable development goals three (SDG3), and to achieve universal health coverage (UHC) [22, 23]. Unfortunately, most efforts by African scientists have been directed towards understanding herbal medicine from biomedical perspectives, consequently excluding spirituality and the totality of traditional medicine from mainstream healthcare options [24–26]. Other related studies focus on the perception of illness by patients [27–31], and exclude the perceptions of the indigenous spiritual healers. Some authors argue for intra-cultural learning to understand and appreciate culturally embedded worldviews [21, 32], and attached meanings.

Traditional spiritual healers may often sound superstitious and are usually only understood by communities of spiritualists. Their healthcare practices, understanding, and descriptions of health, illness, and disease need to be explored further and explained to global audiences even when scientific explanations justifying their clinical practices may be missing [33]. The Uganda national policy guidelines on traditional medicine recommended that, it is important to understand and build consensus about basic definitions, terminologies, and classifications of health, illnesses, and diseases by various sub-categories of traditional medicine to promote information sharing, policy formulations and research methodologies [34], to enable conceptualization of traditional health care practice in various cultures. Despite the national recommendations and policies, there is a paucity of literature that describes health, illness, disease and management approaches as understood by traditional spiritual healers in Uganda. The purpose of the study was to explore the descriptions of health, illness, disease, and health management among Baganda traditional spiritual healers in Central Uganda.

## Method

### Study design and setting

A qualitative and exploratory research design using ethnographic approach was used to explore terminologies that described health, illness, disease and for health management

practices among *Balubaale* in Central Uganda. Uganda area covers 241.555 km$^2$ and is divided into four regions namely Eastern, Western, Northern and Central. It has 156 districts and a population of 44.2 million people of which 88.6% are in rural areas and subsistence farmers [35]. Central region is where the *Balubaale* who engage ancestral spirits during health management are most concentrated in Uganda.

### Sampling and data collection

We identified participants through their governing associations (NACOTHA, Uganda N'eddagala Ly'ayo, Uganda N'eddagala N'obuwangwa Bwaffe) by their leaders [36]. District and cultural leaders completed additional identification. We approached the Healthcare spiritualists at their workplaces (shrines) and invited them to participate. All the remaining twelve (10 Male; 2 Female) participants who agreed and fulfilled the inclusion criteria and signed an informed consent were recruited for the study between 15$^{th}$ July 2019 and 29$^{th}$ April 2020 and were prospectively interacted with for two years. Fig 1 below shows the study profile of the participants.

The inclusion criterion of being a spirit medium for Muwanga affected the need to balance male and female representation. Anybody below 18 years old was excluded. Twelve (10M, 2F) adult healthcare spiritualists (*Balubaale)* residing and working in Central Uganda were purposefully selected [37], based on their background knowledge and at least 10 years of experience. Selected participants were mediums for ancestral spirits including the spirit Muwanga. Related studies successfully used small numbers of participants and retained validity in their studies [36, 38]. Data was collected through semi-structured interviews using open-ended questions and partial integration observation methods. The first author closely worked with the study participants at their workplaces for two years and participated in some daily health care activities. He interviewed the practitioners as appropriate which lessened interference with their usual practices.

### Data management and analysis

The researcher and research assistants directly translated the data into English while preserving the rich context in Luganda, the local language of the spiritualists. Transcribed data was coded and thematically analyzed using ATLAS ti. 22 Computer software based on an inductive approach.

### Ethics consideration

Ethical approval of this study was sought and obtained from the Research and Ethics Committee at Mbarara University of Science and Technology (No. MUREC 1/7) and the Uganda National Council of Science and Technology (SS 4947). The study was also cleared by Traditional Healers' Associations (NACO/0485/2019) and Buganda Kingdom and the Office of the President (ADM 194/212/01). Written informed consent was also obtained from the individual subjects.

## Findings

### Socio-demographic characteristics of respondents

The study participants were twelve (10 Male, 2 Female) Baganda traditional spiritual healers using ancestral spirits in health management (*Balubaale)*. They were aged between 27 and 77 years (Mean age 54), from eight Buganda Counties, eleven Districts, and nine Baganda clans. The majority (66%, n = 8) belonged to traditional religion. Two never attended school, six

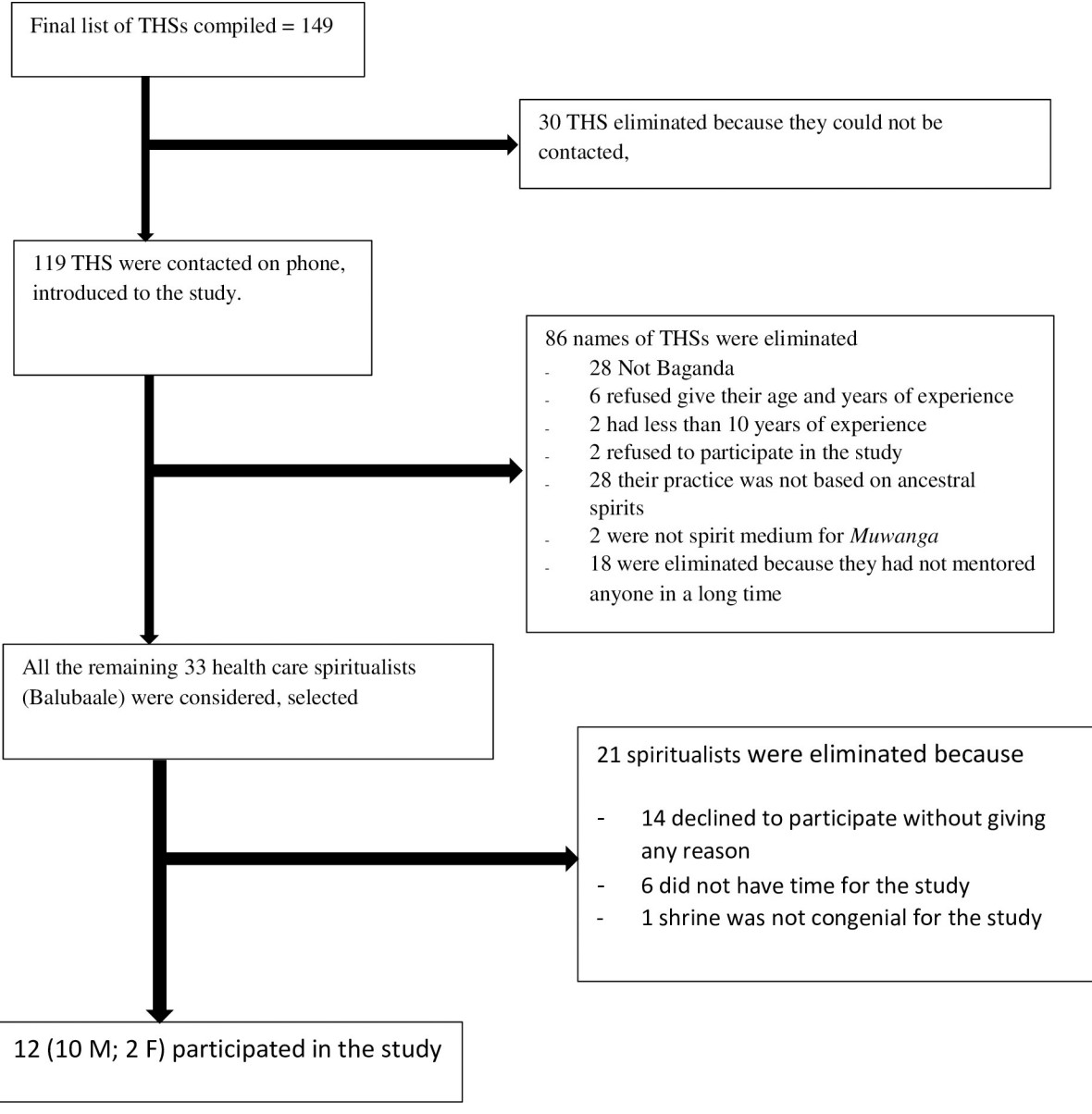

**Fig 1. Study profile.**

never completed primary level, and four entered secondary level. All respondents belonged to healers' associations, had more than ten years of experience, and were subsistence farmers.

## Description of health, illness, and disease among spiritualists

Concepts that described health, illness, and disease among *Balubaale* were embedded in words and phrases used during the various health management approaches. Research findings were based on responses and observed communications of spiritualists while in normal states and as spirits possessed them during health care practices. When the words were analyzed, most of them did not have English equivalent in mainstream medicine. They were left in Luganda, and were given a meaning in English close to the intended meaning in the local language. The words imitated descriptive root causes, therapeutic approaches, and or associated ancestral

spirits. Other words were directly connected with used natural materials such as plants, animals, birds and water sources, influenced by the cultural settings.

Words and phrases used to describe good health varied depending on the state of the spiritualists. Spiritualists used straightforward words and phrases for good health in their normal states, but when they were acting as spirit mediums, the words became more complex.

**Perspectives on good health.** The phrase used to describe good health by the spiritualists, while in their normal states was "*obulamu obulungi*". However, when possessed by ancestral spirits they used the word "*bweeza*". According to the research findings, *obulamu obulungi* (good health) is a complex concept that is bi-dimensional, focusing on physical and nonphysical health.

*Physical and psychological health.* Good health in relation to physical and psychological health includes words that refer to the physical body (*omubiri*), mental (*obwongo*), psychic (*ebirowozo*), and behavior (*empisa*). The spiritualists expressed that much of the manifestations in the physical and psychological reality were mainly at the non-physical level.

*Nonphysical health. Obulamu obulungi (*good health) goes beyond describing the desirable physical human health. It also focuses on the non-physical self, including the state of the soul and the good relations with ancestry. For example, one participant said that "*. . . good health includes aspects of the body, spirit, mind, moral, and soul"* (S4 Data [M, 63]). Furthermore, *obulamu obulungi* focuses on the affordability of basic human needs such as food and shelter for individuals and their family members. Good health was also associated with success in one's endeavors, happiness, joy, a stable family, having children, and good friends. For example, *". . . good health is when a person can afford basic needs like food and housing for self and family"* (S5 Data [M, 64]). The participants made it clear that good health does not mean that one did not get ill or have a disease, but one has the ability to timely access appropriate health care to address their biomedical, social, cultural, mental and spiritual causes of their undesired health. According to the participants, good health was also contributed to by good luck (*omukisa*).

"*Bweeza*", on the other hand, was frequently used by spiritual healers, spirit mediums and ancestral spirits for good health during trance. *Bweeza* is usually aligned with good health and the inner peace that it comes with. For example, *". . . bweeza is a summary word for good health."* (S4 Data [M, 63]), and "*. . . bweeza means, and is associated with peace."* (S9 Data [M, 59]). *Bweeza* was also associated with blessings, fertility, bearing of twins, and rain.

**Perspectives on illness (*olumbe*) and disease (*obulwade*).** Illness was referred to as *olumbe*. Participants described *olumbe* as an unclear and unnamed bad health condition that often occurs suddenly and has many possible root causes. In addition, the person who suffers from *olumbe* sometimes has several complicated medical conditions which are often not cured by mainstream medical approaches. For example, one participant said; *". . . olumbe is something very difficult to understand, is associated with powers of darkness and very difficult to explain."* (S12 Data [M, 27])

Furthermore, participants described *olumbe* in relation to its prospected root-causes, majorly the ancestral spirits (*Lubaale) and witchcraft (ddogo)*. *Lubaale* was considered responsible for causing *olumbe* especially when the ancestral spirits have unfinished business with the person or family. When there are unmet demands, *olumbe* manifests in the physical body, psych, and social status of an individual or family. The participant continued to explain that o*lumbe* due to ancestral spirits may spread into the whole body before it manifests, has hereditary tendencies, and takes a while to treat. One participant said; *"Olumbe attacks the soul and spirit, takes long to be noticed, and may as well be inherited."* (S4 Data [M, 64]).

Nearly all illnesses were attributed to witchcraft. Witchcraft was used to describe the invisible, spiritual, or physical manifestation of illness caused by an individual towards another, using words and materials with spiritual powers, usually out of jealousy. One participant

explained that *"illness may be caused by witchcraft sent by another person out of jealousy."* (S4 Data [M, 27]). *Olumbe* due to witchcraft may affect an individual and the whole family, and if it results in death, the individual dies with it, and such o*lumbe* is often inheritable unless the needful is resolved. Among Baganda, o*lumbe* is expelled from the family by performing rituals of the last funeral rites after someone has died.

On the other hand, the study participants did not express any singular word for disease, but a multitude of signs and symptoms that manifested on the physical and biological body parts such as fever, pain and vomiting. *Obulwadde* takes on the name of the affected body parts, as clarified by one participant *". . . disease affects the body parts or organs and manifests as symptoms of pain in the head, eyes, nose, teeth, liver, kidneys, intestines, heart, lungs, legs, arms, etc."* (S4 Data [F, 61]). In other words, *obulwadde* was not connected to any root causes nor any spirits.

**Differences between illness (*olumbe*) and disease (*obulwadde*).** For the majority of the participants there was a clear distinction between illness and disease in terms of root-causes, effects, and the management processes. During health assessment, illness (*olumbe*) was connected to ancestral spirits and witchcraft as root causes, while diseases (*obulwadde*) were attributed to causes of organisms such as bacteria and worms, and/or physiological malfunctions of the related organs. For the effects, illness (*olumbe*) majorly affected the non-biological parts of humans such as the soul and spirit, while the disease (*obulwadde*) affected the biological body organs and their functions. For example, one participant explained; *"illness only engulfs the body but does not enter into the body organs, while the disease attacks and destroys the body organs"* (S3_Data [F, 61]).

The management process for illness (*olumbe*) was mainly through rituals engaging the use of energies from living and non-living things, symbols, and communication with the ancestrors to prevent and remove the causative agent and protect against other attacks. Participants reported that illness (*olumbe*) is not cured by biomedical treatment, but gets worse instead. The management of disease (*obulwadde*) attacks to kill offending macro/micro organisms and restore the function of the affected organs through treatment using herbs and physical manipulations. One participant explained; *" illness is when the disease gets worse and fails to respond to biomedical treatment and finally leads to death. That is why the last funeral rites are performed to exit the illness out of the deceased's house"* This means that it is the olumbe which actually finally kills a person."* (S10 Data [M, 46]).

Although majority of participants could distinguish between the concepts of illness and disease, some participants did not find it important to differentiate between these concepts. Illness and disease were considered the same. For example, one participant said *"Illness and disease refer to the same thing, they are the same.* (S11 Data [M, 52])

Study participants reported that they used particular words for health assessment and health management depending on context. It was reported that these words and phrases were imbued with the powers and authority of the spiritualists, and were potentiated by the use of specific rituals, prayers, and sacrifices, which were, at times, carried out in specified sacred places and different aspects of the process. Some of these descriptive words cut across the different domains of prevention, protection, treatment, and health promotion depending on context. To best understand these concepts well, we have subdivided them into two sections. The words used in the health assessment process and words used in the health management process.

Words used for the health assessment include *kulagula (*divine/diagnosis*)*, and *kwaaza Lubaale* (cultural ritual process of investigating ancestral spirits) are detailed below.

## Divination (*Kulagula*)

According to the research participants, divination (*kulagula*) describes a cultural process of health assessment and diagnosis among *Balubaale*. It usually focuses on the assessment and

diagnosing of illness, rather that disease. During divination, the spiritual healers may foretell and advise to their clients based on reflective and future foresight of what may or may not be known to the clients. They also support their clients in taking appropriate actions. *Kulagula* is conducted through supernatural means and its purpose is to find out clients' needs, root causes of particular situation, and knowledge and information about future unknown spiritual aspects. One participant said *". . .divination is the process we use to know, inform, foretell, and advise our clients based on reflective and future foresight".* (S4 Data [M, 50])

During divination, the spiritual healers usually use a health assessment tool (*omweso*). The process involves throwing up of shells and beads that have particular meanings and connotations depending on how they fall in relation to one another. The number of times they throw up the shells and beads depend on the nature of the problem and guidance by the invoked ancestral spirits in charge of the divination. The divination process is started by getting in touch with *Muzimu*, (*Mizimu* for plural), the spirit of a dead person, that owns the rest of ancestral spirits, that may manifest and talk through a human spirit medium. Divination is usually followed by exploration of ancestral spirits and related words.

## Exploration of ancestral spirits (*Okwaaza Lubaale*)

*Okwaaza Lubaale* is a cultural ritualistic process of exploration of family and clan ancestral spirits. The prosses is often prompted by illness or disease, engages family and clan members and is led by known and knowledgeable spiritualists *(Ssenkulu)*. It included cleansing rituals, singing, drumming, and calling upon ancestral spirits to express themselves and establish the root cause of illness and disease. One participant stated that *"okwaaza Lubaale is a Baganda cultural practice to explore clan ancestral spirits. It also serves to diagnose and establish the root cause of problems within the family."* (S7 Data [M, 50]).

**Health management process.** The Luganda word that best describes the health management process is *Kuganga*. This measure is taken by the spiritual healers to treat and protect the physical and non-physical bodies, the mind, and the spirit against problems, spiritual attacks, and witchcraft. It also includes protection of family, home, or property, and it is often performed under instructions and guidance of ancestral spirits. *". . . okuganga is a form of power used for protection and prevention against evil spirits and associated problems."* (Participant 6 [F, 65]). *Kuganga* may include ritual cleansing (*kwambulula*), scarification (*kusandaga*), amulets (*ensiriba* and *yirizi*), a concoction of herbs for ritual baths (*kyoogo*), and decoction of herbal mixture for bathing children (*kyogero*).

**Ritual cleansing (*Kwambulula*).** Participants expressed that *kwambulula* is a ritualistic methodological cleansing process involving techniques linked to the spiritual powers of living materials such as plants, animals, and birds, and non-living materials including waters, rocks, and minerals. These materials are used in combination with ancestral spirits and other spiritual powers invoked by the spiritual healer. *Kwambulula* serves to remove undesired spiritual influences from individuals and families, for purposes of protection, prevention, treatment and health promotion. *Kwambulula* is used to cleanse shrines. For example, one participant explained *"Shrines are routinely and ritualistically cleansed of any evil spirits, un-appropriate previous undertakings or witchcraft."* (S2 Data [M, 77])

**Scarification (*Kusandaga*).** Participants described *kusandaga* (scarification) as a health management approach that involves incisions on the physical body with sharp instruments. The freshly made incisions serve as a way of introducing medications into the human body to prevent, protect, and promote good health. Participants believed that medication put on freshly made incisions is more effective when it gets in contact with the human blood while inside the body. Scarification is usually done by a trained spiritual healer, although sometimes,

the ancestral spirits themselves make scarifications to the clients. One participant reported, *". . . at times, the ancestral spirits do scarification to clients at night while they are sleeping."* (S3 Data [F, 61]). However, it was noted that some spirits did not subscribe to rituals involving scarification nor contact with human blood.

**Amulets, (*Nsiriba and Yirizi*)..**   Amulets such as *nsiriba* and *yirize* were described as combinations of small pieces of physical materials with spiritual powers. Physical materials included parts of animals, birds, insects, reptiles, rocks, plants and minerals ritualistically imbued with ancestral spiritual powers. Amulets are tied together and put close to the body for prevention and protection against intentional and non-intentional negative influence of evil spirits and witchcraft. It is common practice to place the amulets in one or more places of the waist, ankle, wrist or upper part of the arm of the individual depending on the instructions given by the spiritual healer or the ancestral spirits.

**The concepts of *Kyoogo* and *Kyogero*.**   *Ekyoogo* was described as a concoction of medicinal herbs, with spiritual values, mixed with water, and are used for ritualistic cleansing and spiritual sanitization of people, especially before acceptance into spiritual space or spiritual activities. For example, one participant described *"Kyoogo is concoction used for cleansing, especially during most spiritual activities. Water type, its content, and timing for ritual cleansing are dependent on purpose of undertaken ritual and guidance by ancestral spirits."* (S1 Data [M, 42])

*Kyogero* was defined as a decoction of medical herbs with spiritual powers used to bath infants and children to cleanse, immunize, prevent, treat, protect, and provide good luck. It is bathed daily, continuously for most of the infants and early childhood years.

## Discussion

The study participants visualized the research question "What words do you use to describe health, illness, and disease?" beyond the bodily physical health to include non-physical aspects of health such as the spirit, soul, and good relations with spiritual, social, cultural, and ancestral life. Illness is attributed to disgruntled ancestral spirits and witchcraft, while disease is attributed to biological and physical damage of the body and its physiological functions. These descriptions are similar to what has been described elsewhere [39, 40], and support the argument that health is not a universal concept [41–43]. Health is variably approached and interpreted by numerous people, cultures, societies, and groups, and there is no single approach that can comprehensively define health in a way that would stand valid and good for all the people, in all communities and places [44]. This is in line with the WHO definition of health, as a state of complete physical, mental, and social well-being and not merely the absence of disease or infirmity, promoted social welfare as an integral component of overall health, and linked health to social environment and living and working conditions [45]. This definition prompts advocating for working, practical, and operational definitions of health, and the adoption of policies and programs for maintaining and improving health [12].

Many authors clearly distinguished disease from illness [31, 40]. Disease relates to concepts of the proper anatomical, physiological, and biological irregularities and dysfunctions of body organs, tissue, and cells [46–48]. This aligns with the universal definition of disease which is based on objective, empirically identifiable, measurable, neutral and value-free pathological issue, engaging microorganisms and health determinants that can be epidemiologically and bio-statistically determined [49, 50].

Illness, on the other hand, is perceived as personal and interpersonal suffering based on social-cultural standards and values [3, 51, 52], with a multitude of signs and symptoms that may or may not manifest in the human body [40]. Illness is philosophically based on mystical concepts [39, 53], which are usually culturally structured [41, 54, 55], and attached with

various social meanings [56, 57]. Some authors documented illness as an unclear and unnamed spiritual health condition caused by ancestral spirits and witchcraft [39], whose description of illness is embedded in people's cultures. Other authors assert that illness is associated with misfortunes [58, 59] and somatic experiences reflective of bad relations with supernatural forces and ancestral spirits [60]. Illness has various possible causes that may include, but are not limited to curses, witchcraft, and failure to observe social taboos [61], while others indicate that illness may manifest as a divine punishment [41, 62].

In contrast, some authors expressed that illness and disease were in perspectives of first person and third person, where illness is the first person experience and feeling of the condition [47, 63], while disease is the third person's perspective and knowledge of somebody's medical condition [63, 64]. However, Inthorn expressed the same but differently, when he said that illness is the subjective understanding of the lack of health, while the perspective of medical professionals on medical conditions is the disease [65, 66]. They, nonetheless, added that illness and disease are not the same, since a person can feel well and at the same time have an undiscovered tumor, or a person can feel sick without a medical condition [65].

Another important aspect is the concept of health assessment that involves divination and ritual cleansing to establish the health status [67], the prospected root cause, the appropriate management approaches, and prevention [3, 39, 68]. For example, traditional healers in North Eastern Ethiopia use divination to establish the etiology of illness [69]. Other healers use divination to distinguish between illness of supernatural origin and illness described as normal or natural [70–72]. In contrast with the western medicine approach where health status and disease are assessed with physical examination involving inspection, palpitation, percussion, and auscultation [73, 74], illness claims are unpredictable, lack scientific rigor and its measures are neither clear nor objective [75].

Another example is ritual cleansing which aligns with the cultural practices of the Swati traditional healers in Mpumalanga Province, South Africa who use ritual cleansing to manage depression and mental illness [76]. In a related case, in Peru, Ayahuasca rituals are used to cleanse and understand the meaning of the illness and establish a balanced relationship with illness and treatment [77]. In Indonesia, traditional rituals are used to reduce anxiety and build better mental health in response to natural and non-natural illness [78], while in Northern Uganda, Mwaka reported on spiritual healers who use diagnostic rituals to establish spiritual causes of illness [3].

In their comprehensive reviews, Ayeni et al., and Uzobo et al., reported that scarification in Africa was a cultural process used to manage chronic health conditions such as epilepsy, sarcoidosis, and psoriasis [79, 80]. In a recent study by Yuping Zheng et al. scarification is described as a pathological process that stimulates chemicals within the transmembrane proteins to indirectly activate immune cells to release cytokines and neuropeptides for the body's immune defense mechanism for prevention and treatment [81]. However, scarification has been discouraged because of its potential to transmit hepatitis B and human immunodeficiency virus (HIV) [82, 83]. Similarly, the use of amulets, the locally made symbolic ritual bands of power, has been reported to offer protection and healing from spiritual, psychological social, and cultural undesired forces [84]

## Study limitations

Although this study was conducted at multiple healers' shrines (sites), it was limited to one tribe, Baganda, and one category of traditional spiritual healers (*Balubaale*). This limited the breadth of the data as other categories of traditional healers could have provided different perspectives or relevant information to the research question. Another limitation is that the

study-area was limited to Central Uganda. Possibly, if the study area was extended beyond the Central Region, more insightful conclusions could have been generated. The small number of 12 participants was also a limitation of the study. These limitations restrict the generalizability of the study findings, even though the researcher closely worked with the research participants for two years, which enabled the researcher to access sacred places and privileged information.

## Conclusion and recommendation

The research findings show that although the concepts of health, illness, disease are universally used and are assumed to mean the same thing, Baganda traditional spiritual healers *(Balu-baale)* use them uniquely and differently. These words and the health management methods and approaches have a cultural context and are descriptive of the nature of the problem, prospected root causes, therapeutic interventions, and associated spiritual entities. These findings are significant as they contribute to literature in this field and must be recognized and valued for policy considerations in order to build and empower ancestral knowledge and healing.

We recommend more studies in other ethnic or tribal groupings, to explore words used to describe health, illness, disease and related management approaches so as to compare results and guide inter-medical systems communication and health policy formulations.

## Supporting information

**S1 Data. Study participant 1.**
(PDF)

**S2 Data. Study participant 2.**
(PDF)

**S3 Data. Study participant 3.**
(PDF)

**S4 Data. Study participant 4.**
(PDF)

**S5 Data. Study participant 5.**
(PDF)

**S6 Data. Study participant 6.**
(PDF)

**S7 Data. Study participant 7.**
(PDF)

**S8 Data. Study participant 8.**
(PDF)

**S9 Data. Study participant 9.**
(PDF)

**S10 Data. Study participant 10.**
(PDF)

**S11 Data. Study participant 11.**
(PDF)

**S12 Data. Study participant 12.**
(PDF)

**S1 Text. Contextual definitions (legends).**
(DOCX)

## Acknowledgments

We are indebted to the leadership of Mbarara University and the Pharmbiotrac program for soliciting the financial support for the first author PhD study, in particular, Dr. Casim Tolo, Professor Patrick Ogwang, and Engineer Anke Weisheit. We extend our utmost gratitude to the study participants and their spiritual guides for information, free atmosphere, and time extended to the research process. We are grateful to the leaders of traditional healers' associations, namely Sylvia Namutebi (Maama Fiina) of Uganda N'eddagala Ly'ayo N'obuwangwa Bwaffe, Musasizi Karim of NACOTHA, and Karim Walyabira of Uganda N'eddagala Ly'ayo. We are thankful to Buganda Kingdom administration for introducing us to the clan leaders whose support made the interaction with the spiritualists smooth. We thank the research assistants Mr. Eric Kibirige Mukasa and Ms. Nambuya Barbra for the field work. We also thank Dr. John Chrysostom Katongole of Munyonyo Botanical Gardens, Jovent K. Obbo of Bugema University, and Dr. Kizito Simon of Makerere University for their comments on the draft report.

## Author Contributions

**Conceptualization:** Yahaya H. K. Sekagya, Charles Muchunguzi, Payyappallimana Unnikrishnan, Edgar M. Mulogo.

**Data curation:** Yahaya H. K. Sekagya.

**Formal analysis:** Yahaya H. K. Sekagya, Charles Muchunguzi, Edgar M. Mulogo.

**Investigation:** Yahaya H. K. Sekagya.

**Methodology:** Yahaya H. K. Sekagya, Charles Muchunguzi, Payyappallimana Unnikrishnan, Edgar M. Mulogo.

**Supervision:** Charles Muchunguzi, Payyappallimana Unnikrishnan, Edgar M. Mulogo.

**Validation:** Charles Muchunguzi, Payyappallimana Unnikrishnan, Edgar M. Mulogo.

**Visualization:** Yahaya H. K. Sekagya, Charles Muchunguzi, Payyappallimana Unnikrishnan, Edgar M. Mulogo.

**Writing – original draft:** Yahaya H. K. Sekagya, Charles Muchunguzi, Payyappallimana Unnikrishnan, Edgar M. Mulogo.

**Writing – review & editing:** Yahaya H. K. Sekagya, Charles Muchunguzi, Payyappallimana Unnikrishnan, Edgar M. Mulogo.

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
