## [Decision Letter · Decision Letter 0]

11 Mar 2024

PGPH-D-23-01744

Perspectives on health, illness, disease and management approaches among traditional health care spiritualists in Central Uganda.

Dear Dr. Sekagya,

Thank you for submitting your manuscript to PLOS Global Public Health. After careful consideration, we feel that it has merit but does not fully meet PLOS Global Public Health’s publication criteria as it currently stands. Therefore, we invite you to submit a revised version of the manuscript that addresses the points raised during the review process.

We look forward to receiving your revised manuscript.

Kind regards,

Gideon Lasco

Academic Editor

Journal Requirements:

Additional Editor Comments (if provided):

Dear authors,

As you can see, both reviewers find merit to the article, but they find the need to relate it with the existing literature on traditional healing in Uganda and the broader region - and also to bring out more clearly what this study adds to the scholarship. I would like to give you the opportunity to revise the paper by closely engaging with their constructive comments.

Reviewers' comments:

Reviewer's Responses to Questions

**Comments to the Author**

1. Does this manuscript meet PLOS Global Public Health’s publication criteria? Is the manuscript technically sound, and do the data support the conclusions? The manuscript must describe methodologically and ethically rigorous research with conclusions that are appropriately drawn based on the data presented.

Reviewer #1: No

Reviewer #2: Yes

2. Has the statistical analysis been performed appropriately and rigorously?

Reviewer #1: No

Reviewer #2: No

3. Have the authors made all data underlying the findings in their manuscript fully available (please refer to the Data Availability Statement at the start of the manuscript PDF file)?

Reviewer #1: Yes

Reviewer #2: Yes

4. Is the manuscript presented in an intelligible fashion and written in standard English?

Reviewer #1: No

Reviewer #2: No

5. Review Comments to the Author

Reviewer #1: You don't need the acronym THCS. Just call them "traditional healers" as they are referred to everywhere else in the literature. The acronym is too cumbersome.

Regarding the third, fourth, and fifth sentence in the Introduction: These are oral cultures and are highly stigmatised especially by locals (viewed as "evil"/witchcraft). These are the primary reasons there is little research on this subject.

If you're going to use a term as loaded as "knowledge apartheid" you're going to need to do much more explaining as to what the means and why that is appropriate.

Traditional medicine is not used by high percentages of Americans, Australians, Germans and Canadians. "Traditional healing" as you are examining it is directly related to pre-Christian and pre-Islamic African traditional religions. They are still present today in a syncretic form with Christianity and Islam. This is why they exist so significantly in Africa and not in the places you mention. You need to do significantly more reading on the background of traditional religion in Africa as the authors do not seem to grasp the socio-religious context of the subject they are treating here. This is very concerning.

Overall the Introduction has serious problems. More reading should be done about studies of traditional healing in other countries in east and southern Africa (which share many of the same traditional spiritual belief systems and practices related to traditional healing).

I'm concerned about the disproportionate numbers of men interviewed compared to women. There needs to be a more thorough justification of this imbalance. How did this affect the findings?

There is no need to capitalise words like "Physical" or "Physiological" or "Good Health". There are also sloppy mistakes with italics. Overall the text needs to be fully reviewed for spelling, grammar and punctuation more thoroughly.

The Results section is difficult to follow. Why are all these terms important? There are also serious problems with translation into English. For example, the dichotomy you present about "illness" versus "disease" should actually be translated as "supernatural illness" (connected to ancestral spirits and witchcraft) and "natural illness" (connected to biomedical problems). Many other studies have documented this distinction throughout east and southern Africa. Use terms already established in the literature to avoid confusion.

Quotes in the Divination section should be incorporated into the text. Not listed at the end. Why are these quotes important? Similar divination practices have been documented widely throughout east and southern Africa. The authors should fully research these examples. This is the same with ritual cleansing and scarification (often referred to as "cutting" in the literature).

Overall, the authors need to do significant more research about traditional healer practices already documented throughout east and southern Africa. The authors seem to think their study is the first (or one of the first) on this subject. This is incorrect. There are many studies that describe similar rituals, practices and concepts and the authors don't seem to have incorporated any of these ideas into their text. This is a serious concern. After more reading and research about previous studies have been completed by the authors, there needs to be an extensive revision of the text.

Reviewer #2: • The article is really interesting. This article explores how traditional health care spiritualists in central Uganda see health, illness, disease, and management practices.

• Title: The title explains the purpose and objective of the article.

• Abstract: The abstract offers brief summary of the paper, and the language used in the abstract is easy to read and understand.

Introduction:

• Introduction: The introduction should include three parts, which are the background on the topic, the significance of the topic, and the aim of the study. The introduction covered these three parts to some extent.

• Correction: introduced education curricula banned people using...replace with: introduction of education curricula banned people from using...

• Correction: in relation to concepts ...replace with: about concepts...

• Correction: insistence should be for traditional ...replace with: emphasis should be on traditional...

Methods:

• The authors mentioned that Uganda is divided into four regions, while the study covered only the central part. The authors did not explain the reason behind choosing this particular region to conduct the study. So, please explain the reasons for more clarity.

• Correction: Additional identification was done by district and cultural leaders...replace with: District and cultural leaders completed additional identification....

• It has been mentioned that the first researcher was very close to the participants and worked with them in their workplaces for two years, while the specific period of the study was approximately one year (between 15th July 2019 and 29th April 2020)...So, please explain that for more clarity.

• Please separate the inclusion criteria from the exclusion criteria and explain them adequately. It is noted that the focus has been on the inclusion criteria and merging them with the exclusion criteria, so please separate them and explain them sufficiently for greater clarity.

• Correction: The data was transcribed by the researcher and research assistants directly into English while maintaining depth of context in Luganda, the local language of spiritualists...replace with: The researcher and research assistants directly translated the data into English while preserving the rich context in Luganda, the local language of spiritualists.....

• Although the study consumed a great deal of effort from researchers to deal with spiritual healers and understand the rituals used in addition to translating terminology, the number of participants is very small to explore this issue, especially since there is a difference in some participants’ opinions and explanations regarding some terminology.

FINDINGS:

• Correction: Study participants were twelve (10 Male, 2 Female) ...replace with: The study participants were twelve (10 males and 2 females).....

• It has been mentioned that most words did not have English equivalents in mainstream medicine. So, when they were analyzed, did they give a meaning close to the intended meaning in the local language?

Discussion:

• Please shorten the first paragraph and focus on discussing the results.

• The discussion needs to be supported by additional references to compare the results of the current study with those of previous studies. The discussion focused on comparison with the results of two previous studies from Ethiopia and Saudi Arabia. Therefore, I suggest that the authors support the discussion section with references and other recent studies, especially from developing countries, which rely heavily on this type of medicine.

Conclusion:

• The research offers sample data for the authors to draw conclusions from.

Grammar: Need revision along the manuscript.

6. PLOS authors have the option to publish the peer review history of their article (what does this mean?). If published, this will include your full peer review and any attached files.

**Do you want your identity to be public for this peer review?** For information about this choice, including consent withdrawal, please see our Privacy Policy.

Reviewer #1: No

Reviewer #2: No

---

## [Decision Letter · Decision Letter 1]

7 May 2024

PGPH-D-23-01744R1

Perspectives on health, illness, disease and management approaches among Baganda traditional spiritual healers in Central Uganda.

Dear Dr. Sekagya,

Thank you for submitting your manuscript to PLOS Global Public Health. After careful consideration, we feel that it has merit but does not fully meet PLOS Global Public Health’s publication criteria as it currently stands. Therefore, we invite you to submit a revised version of the manuscript that addresses the points raised during the review process.

We look forward to receiving your revised manuscript.

Kind regards,

Gideon Lasco

Academic Editor

Journal Requirements:

2. Please provide separate figure files in .tif or .eps format only and remove any figures embedded in your manuscript file. Please also ensure all files are under our size limit of 10MB.

Additional Editor Comments (if provided):

Thank you for the revision! As you can see, both reviewers are positive about the changes done, but Reviewer 1 has very detailed comments for further strengthening the paper and I encourage you to revisit the text in light of them.

Reviewers' comments:

Reviewer's Responses to Questions

**Comments to the Author**

1. If the authors have adequately addressed your comments raised in a previous round of review and you feel that this manuscript is now acceptable for publication, you may indicate that here to bypass the “Comments to the Author” section, enter your conflict of interest statement in the “Confidential to Editor” section, and submit your "Accept" recommendation.

Reviewer #1: (No Response)

Reviewer #2: All comments have been addressed

2. Does this manuscript meet PLOS Global Public Health’s publication criteria? Is the manuscript technically sound, and do the data support the conclusions? The manuscript must describe methodologically and ethically rigorous research with conclusions that are appropriately drawn based on the data presented.

Reviewer #1: Yes

Reviewer #2: Yes

3. Has the statistical analysis been performed appropriately and rigorously?

Reviewer #1: N/A

Reviewer #2: Yes

4. Have the authors made all data underlying the findings in their manuscript fully available (please refer to the Data Availability Statement at the start of the manuscript PDF file)?

Reviewer #1: Yes

Reviewer #2: Yes

5. Is the manuscript presented in an intelligible fashion and written in standard English?

Reviewer #1: Yes

Reviewer #2: Yes

6. Review Comments to the Author

Reviewer #1: Delete the third phrase in the introduction or clarify what “colonizing ideology” means. While Christianity 100 years ago did demonize traditional religious practices, it continues to do so today almost exclusively at the behest of “born again” locals. This is perhaps the largest reason as to why traditional healing stays in the shadows to such a large extent today as those practicing are it persecuted by their fellow Ugandans. It is a taboo subject in Ugandan society and remains difficult to discuss much less write about. This should be noted.

Avoid using any language that shows the author’s opinions or personal value judgments: “unethical” in paragraph 1. Also what does paragraph 2 in the introduction mean? Healers have had multiple accusations? About what? By whom? And they’ve “lost pride and confidence”? In what? If they didn’t still have confidence, they wouldn’t be doing this work so that sentence makes no sense. There remain sentences that should be reviewed by a native speaker to ensure overall readability.

The authors have chosen to continue creating a distinction between "illness" and "disease" (olumbe and obulwadde in the text). This is a mistake as it doesn't make sense in English. If authors do not like the distinction of "natural" versus "supernatural" illness they need to find a better way to distinguish them in English as "illness" and "disease" are synonomous and should not be juxtaposed. The literature (including many articles cited below) largely utilizes the distinction of "natural" and "supernatural" illnesses and I strongly suggest the authors use either those terms of terms already cited in the literature to highlight these differences. "Illness" and "disease" are unacceptable.

Authors should review the literature below and include in the Introduction and Discussion to improve overall analysis and context of study:

Abbo, C., Okello, E. S., Musisi, S., Waako, P., & Ekblad, S. (2012). Naturalistic outcome of treatment of psychosis by traditional healers in Jinja and Iganga districts, Eastern Uganda–a 3-and 6 months follow up. International Journal of Mental Health Systems, 6(1), 1-11.

Galvin, M., Chiwaye, L., Moolla, A. (2023d). Religious and Medical Pluralism among Traditional Healers in Johannesburg, South Africa. Journal of Religion and Health, 1-17. doi: 10.1007/s10943-023-01795-7

Galvin, M., Chiwaye, L., Moolla, A. (2023e). Perceptions of Causes and Treatment of Mental Illness Among Traditional Health Practitioners in Johannesburg, South Africa. South African Journal of Psychology, 1-13. doi: 10.1177/00812463231186264

Louw, G., & Duvenhage, A. (2016). The present-day diagnosis and treatment model of the South African traditional healer. Australasian Medical Journal.

Peltzer, K. (1999). Faith healing for mental and social disorders in the Northern Province (South Africa). Journal of Religion in Africa, 29(Fasc. 3), 387–402.

Sorsdahl, K. R., Flisher, A. J., Wilson, Z., & Stein, D. J. (2010a). Explanatory models of mental disorders and treatment practices among traditional healers in Mpumulanga, South Africa. African Journal of Psychiatry, 13(4), 284–290. https://doi.org/10.4314/ajpsy.v13i4.61878

Sorsdahl, K., Stein, D. J., & Flisher, A. J. (2010b). Traditional healer attitudes and beliefs regarding referral of the mentally ill to Western doctors in South Africa. Transcultural Psychiatry, 47(4), 591–609. https://doi.org/10.1177/1363461510383330

Reviewer #2: I would like to express my thanks to the authors for their excellent work in responding to the reviewers' remarks. The authors have made all necessary changes and implemented all previous recommendations

7. PLOS authors have the option to publish the peer review history of their article (what does this mean?). If published, this will include your full peer review and any attached files.

**Do you want your identity to be public for this peer review?** For information about this choice, including consent withdrawal, please see our Privacy Policy.

Reviewer #1: No

Reviewer #2: No

---

## [Editor Report · Decision Letter 2]

2 Jul 2024

PGPH-D-23-01744R2

Perspectives on health, illness, disease and management approaches among Baganda traditional spiritual healers in Central Uganda.

Dear Dr. Sekagya,

Thank you for submitting your manuscript to PLOS Global Public Health. After careful consideration, we feel that it has merit but does not fully meet PLOS Global Public Health’s publication criteria as it currently stands. Therefore, we invite you to submit a revised version of the manuscript that addresses the points raised during the review process.

We look forward to receiving your revised manuscript.

Kind regards,

Gideon Lasco

Academic Editor

Journal Requirements:

2. We have noticed that you have uploaded Supporting Information files, but you have not included a list of legends. Please add a full list of legends for your Supporting Information files after the references list.

Additional Editor Comments (if provided):

Dear Authors,

Thank you for your efforts to revise the paper. I believe that we are nearly ready to accept the paper, but in terms of overall form, the manuscript needs editing and sharpening of the language as there are some passages that are not clear. For instance (the following are illustrative but not exhaustive):

p. 3 - The worldviews of traditional spiritual healers regarding health, illness, disease, and their management approaches are rooted in their local oral cultures and traditions, are highly stigmatized as evil and are associated with witchcraft. During colonial period, indigenous people were described as primitive and their knowledge systems rendered obsolete, irrelevant and worthless (Use of passive voice: Highly stigmatized by whom? Described as primitive by whom?)

p. 4 - Although, nearly 80% of the populations in developing countries use traditional medicine [11,20], it should be noted that indigenous spiritual healers and their use of local descriptive words during health management are not acknowledged [7,21]. (Again, the passive voice raises questions: Not acknowledged by who?)

p. 4 - Unfortunately, most efforts have been directed towards understanding herbal medicine from biomedical perspectives, consequently excluding spirituality and the totality of traditional medicine from mainstream health care options (Most efforts by who?)

p. 15 - The main findings to the research question “what words do you use to describe health, illness and disease?” were visualized beyond the bodily physical health to include nonphysical aspects of health such as the spirit, soul and good relations with spiritual, social, cultural and ancestral life. (Use of passive voice, not clear what the authors are referring to)

p. 15-16 - The discussion on disease vs. illness is not well justified in relation to the findings

Please carefully read through the text for any errors including spelling and grammar, particularly in the introduction and discussion, using the passive voice only when called for, and taking great care in avoiding vague sentences and making sure that readers understand who is performing the actions mentioned, or being referred to.

Thank you - we look forward to receiving the tweaked manuscript.
---

## [Editor Report · Decision Letter 3]

12 Aug 2024

Perspectives on health, illness, disease and management approaches among Baganda traditional spiritual healers in Central Uganda.

PGPH-D-23-01744R3

Dear Dr. Sekagya,

We are pleased to inform you that your manuscript 'Perspectives on health, illness, disease and management approaches among Baganda traditional spiritual healers in Central Uganda.' has been provisionally accepted for publication in PLOS Global Public Health.

Best regards,

Gideon Lasco

Academic Editor

Thank you for your patience in the review process and for addressing the editorial concerns raised; we are pleased to accept your paper.